# An Integrated Smart Pond Water Quality Monitoring and Fish Farming Recommendation Aquabot System

**DOI:** 10.3390/s24113682

**Published:** 2024-06-06

**Authors:** Md. Moniruzzaman Hemal, Atiqur Rahman, Farhana Islam, Samsuddin Ahmed, M. Shamim Kaiser, Muhammad Raisuddin Ahmed

**Affiliations:** 1Department of IoT and Robotics Engineering, Bangabandhu Sheikh Mujibur Rahman Digital University, Bangladesh, Kaliakair, Gazipur 1750, Bangladesh; 1801007@iot.bdu.ac.bd (M.M.H.); 1801025@iot.bdu.ac.bd (A.R.); samsuddin0001@bdu.ac.bd (S.A.); 2Division of Computer Science and Engineering, Louisiana State University and Agricultural and Mechanical College, Baton Rouge, LA 70803, USA; 3Department of Educational Technology, Bangabandhu Sheikh Mujibur Rahman Digital University, Bangladesh, Kaliakair, Gazipur 1750, Bangladesh; farhana0001@bdu.ac.bd; 4Institute of Information Technology, Jahangirnagar University, Savar, Dhaka 1342, Bangladesh; mskaiser@juniv.edu; 5Radio and Radar Communication, Military Technological College, Muscat 111, Oman; muhammad.ahmed@mtc.edu.om

**Keywords:** smart fish farming, Internet of Things (IoT), smart robot, machine learning, solar power, real-time water quality

## Abstract

The integration of cutting-edge technologies such as the Internet of Things (IoT), robotics, and machine learning (ML) has the potential to significantly enhance the productivity and profitability of traditional fish farming. Farmers using traditional fish farming methods incur enormous economic costs owing to labor-intensive schedule monitoring and care, illnesses, and sudden fish deaths. Another ongoing issue is automated fish species recommendation based on water quality. On the one hand, the effective monitoring of abrupt changes in water quality may minimize the daily operating costs and boost fish productivity, while an accurate automatic fish recommender may aid the farmer in selecting profitable fish species for farming. In this paper, we present AquaBot, an IoT-based system that can automatically collect, monitor, and evaluate the water quality and recommend appropriate fish to farm depending on the values of various water quality indicators. A mobile robot has been designed to collect parameter values such as the pH, temperature, and turbidity from all around the pond. To facilitate monitoring, we have developed web and mobile interfaces. For the analysis and recommendation of suitable fish based on water quality, we have trained and tested several ML algorithms, such as the proposed custom ensemble model, random forest (RF), support vector machine (SVM), decision tree (DT), K-nearest neighbor (KNN), logistic regression (LR), bagging, boosting, and stacking, on a real-time pond water dataset. The dataset has been preprocessed with feature scaling and dataset balancing. We have evaluated the algorithms based on several performance metrics. In our experiment, our proposed ensemble model has delivered the best result, with 94% accuracy, 94% precision, 94% recall, a 94% F1-score, 93% MCC, and the best AUC score for multi-class classification. Finally, we have deployed the best-performing model in a web interface to provide cultivators with recommendations for suitable fish farming. Our proposed system is projected to not only boost production and save money but also reduce the time and intensity of the producer’s manual labor.

## 1. Introduction

The need for protein is rising daily in tandem with the expanding population. Fish is a popular protein source that is low in fat and rich in critical micronutrients that are vital to human health. However, the current fish farming method seems to be insufficient to meet the anticipated demand of the expanding population [1]. The amount and quality of fish produced in fish farming are greatly influenced by the state of the water in which they live. However, the growing solid and hazardous waste disposal in water sources, as well as climate change, are lowering the water quality and significantly harming aquatic life [2,3]. As a result, pond fish farmers must pay attention to crucial components of the pond’s water on a regular basis [4]. The physical, chemical, and biological properties of pond water have a substantial impact on fish production. As a result, smart fish farming approaches necessitate monitoring and managing all of these attributes. Temperature, pH, turbidity, dissolved oxygen (DO), total dissolved solids (TDS), water color, carbon dioxide, alkalinity, electrical conductivity, unionized ammonia, nitrate, nitrite, and other parameters related to water quality are all useful [5,6,7,8]. Furthermore, sudden changes in water, as well as filthy water, cause a range of fish illnesses, including death. If the quality of the pond water is not within the appropriate range, fish farmers can replace or change the water to ensure fish development and improve the water quality. By monitoring the levels of these factors, they may make timely decisions to change the water; otherwise, water replacement might be exceedingly harmful. The prior approaches still depend on human effort in testing to determine parameter values, which is deemed to be less effective because parameter values change with time [9]. Moreover, sudden changes in water, as well as polluted water, are responsible for various diseases in fish and even death. To ensure the growth of fish and to improve the quality of water, fish farmers can replace or modify the pond water if the quality is not within the required range. While observing the values of these parameters, they can make timely decisions to change the water; otherwise, the replacement of water can be highly dangerous.

The traditional system still depends on manual testing to determine the parameter values, which is considered less effective because the parameter values may change over time. Furthermore, this procedure is time-consuming, labor-intensive, and costly. Data-driven decisions can increase productivity by minimizing sudden losses [10]. Designing and constructing a real-time water surveillance system with a data-driven decision-making capacity might aid in mitigating these issues [11,12,13]. To address this issue, state-of-the-art technologies may be used to increase productivity and minimize losses through the continuous monitoring of water quality indicators and prompt decisions.

In this paper, we present an IoT-enabled cloud-based system that can monitor water quality, make data-driven choices, and recommend suitable fish species for farming. A solar-powered IoT-enabled robot is designed to provide real-time water quality data. The robot collects temperature, pH, and turbidity data using wireless sensors, which are then processed by a microcontroller and transmitted to cloud-based platforms through ESP-32. The data are then captured and analyzed in real time, allowing fish farmers to receive real-time updates from anywhere and make informed decisions based on the findings. To record the data, Google Sheets and Firebase are utilized. Using solar panels as a substitute source of electrical power reduces the power usage significantly, making the system more economical and self-sustaining. In addition to this, the robot uses very few sensors, making it more cost-effective and user-friendly. This study also contributes to developing a web-based fish recommendation system that uses machine learning algorithms to assess the data saved in Google Sheets and provide recommendations for suitable fish species for farming. This system assists the fish farmer in selecting acceptable fish for cultivation based on the pond’s water quality characteristics, enhancing production and profitability. The main contributions of this study can be summarized as follows:Development of an IoT-enabled cloud-based water monitoring system that provides real-time water quality data and allows access to these data from any location;Implementation of a mobility-based single-sink data gathering approach, enabling the temporal interval sampling of real-time data from sensors;Integration of proposed ensemble method to analyze the data stored in a cloud database and generate suggestions for suitable fish species based on the analysis;Creation of a user-friendly web interface that enables users from anywhere to identify appropriate fish species based on three factors—temperature, pH, and turbidity;Utilization of solar panels as power sources, making the system self-powered, sustainable, cost-effective, and highly autonomous.

The rest of the paper is organized as follows. Section 2 describes recent related works. Section 3 provides an in-depth analysis of the approach and methodology. The experimental data and analyses are presented in Section 4. The comparison of several state-of-the-art methods with our proposed model is described in Section 5. Finally, Section 6 contains the conclusions of the work.

## 2. Literature Review

The Internet of Things (IoT) and machine learning are becoming increasingly important in fish farming, particularly in the development of real-time water quality monitoring systems. Several studies [10,14,15,16,17,18,19,20,21,22,23,24] have previously been undertaken on the benefits of this technology in this specific field. Some studies focus on monitoring and surveillance, while others are more concerned with fish species recommendations. In this section, we have included a brief summary of the related works.

Raju et al. [10] proposed an IoT-based water quality monitoring system that collects quality data by using multiple sensors and transmits the collected data to a cloud database for analysis. The proposed method uses a Raspberry Pi with an in-built Wi-Fi module as a microcontroller, which works as the system’s internet gateway. If the quality data deviate from the optimum ranges, a message with a feasible solution will be sent to the aqua farmer device through a mobile app. They implemented a water quality monitoring system using IoT and cloud systems, but they did not design a recommender system for the selection of appropriate fish based on the values of the water quality indicators.

Cordova-Rozas et al. [14] developed a cloud-based water monitoring system. It is based on five phases: data extraction, cloud platform, database storage, reports, and predictions. First, data are extracted using sensors and a microcontroller. The data are then transferred to the cloud platform and saved in a cloud database. The recorded data are then shown on dashboard reports. Finally, based on the previous data, a prediction for each water parameter is displayed. They did not incorporate an alternative renewable power source or design a recommender system.

Gao et al. [15] developed an IoT-based fish farming and tracking control system consisting of an intelligent management module and an aquatic product tracking module. The intelligent management module gathers water quality parameter values using integrated sensors and achieves the continuous monitoring, storage, analysis, and forecasting of water quality data. The aquatic product tracking module provides consumers with farming information such as water quality, geographical location, and transportation through a QR code tag attached to the product. The authors did not consider suggesting any particular fish species for farming and instead concentrated on the intelligent control and management of fishpond equipment.

Nagayo et al. [16] developed a solar-powered aquaponics system incorporating a water-recirculating system, an Arduino-based and Labview-based aquaponics control and monitoring system, and a cooling and heating system that balances the temperature for plant and fish growth. Despite the fact that they have presented a multidisciplinary strategy for the monitoring and control of water quality, the system lacks a decision support capacity.

Pasika et al. [18] suggested a low-cost, high-efficiency water quality monitoring system that consists of many sensors that sense different parameters, such as the pH, turbidity, the water level in the tank, the temperature, and the humidity of the surrounding atmosphere. The collected data are transferred to the cloud via the IoT-based ThinkSpeak application. This study also did not include components of decision support for fish farmers.

Huan et al. [19] proposed a water quality monitoring system based on narrow-band IoT for aquaculture ponds. It uses multiple sensors to collect pond water parameters (such as pH, temperature, and dissolved oxygen) and then sends the data reports to the cloud platform via the NB module. NB-IoT offers real-time, reliable data acquisition and quickly responds to commands, thus maximizing aquaculture productivity. Despite the fact that the system has stable overall operation and can meet current production needs while also providing strong information and support services for future water quality control and aquaculture production management, it does not have automated decision features.

Chiu et al. [20] designed an IoT-based aquaculture monitoring and control system to collect and analyze real-time water quality data. The authors also developed a deep learning model that correlates the collected water quality parameters and anticipates the growth of the California bass fish. The approach focused on feeding rather than fish species recommendations.

Niswar et al. [25] presented a water quality monitoring system for crab farming based on IoT technology to improve softshell crab’s survival and productivity. While the system incorporates MQTT and a LoRa-based wireless sensor network for communication with small embedded devices, mobile devices, and sensors and has a web-based monitoring platform for remote access to water quality data, it does not possess intelligent decision-making capabilities or recommendation features.

Pikulins et al. [26] developed the software and hardware required for a self-sustaining water quality monitoring system (WQMS) to track water quality. The system is equipped with essential hardware and software components to collect data from remote areas of the pond. It offers storage, data processing, and the control of active devices. However, it does not have an automated decision support feature.

Murad et al. [27] designed a low-cost, solar-powered aquaponics system including pH and temperature sensors, a water sensor, a servo, an LCD, a peristaltic pump, solar power, and GSM to monitor and adjust the water parameters. The collected information is presented on an LCD screen, and if the sensor readings are abnormal, a text message is delivered over GSM to the user’s phone. This system also has limitations due to the intelligent components.

Abinaya et al. [28] proposed an IoT-based water monitoring system composed of multiple sensors, an Arduino, and a GSM module to measure the temperature, pH, DO, water level, ammonia, and foul smells. If the measured values depart from the ideal range, the Arduino activates the associated controller to undertake the necessary steps, and a message is sent to the concerned individual through the GSM module. This system also lacks intelligent components.

Islam et al. [29] proposed a model to predict the appropriate fish species based on parameters such as the pH, temperature, and turbidity in an aquatic environment. They implemented J48, RF, kNN, and CART on a real-time pond water dataset for fish farming and achieved 88.48% accuracy, an 87.11% kappa statistic, and an 88.5% TP rate with the RF model.

The majority of existing systems [10,14,15,18,19,25,26,27,28] lack automated decision support features. A promising avenue for research involves the integration of machine learning or AI algorithms to furnish intelligent recommendations for fish farming based on real-time water quality data. Notably, none of the referenced studies [10,14,15,20,25,27] specifically design a recommender system for the selection of suitable fish species based on water quality indicators. Furthermore, a limited number of studies incorporate alternative renewable power sources in their monitoring systems [14,15]. It would be beneficial for research to explore the integration of renewable energy sources, enhance sustainability, and reduce the dependence on traditional power sources. While certain studies [15,26] present multidisciplinary strategies, a notable gap exists regarding the absence of comprehensive approaches that integrate various facets, such as water quality monitoring, renewable energy, and fish farming recommendations. Additionally, some studies [14,19,20,25] lack real-time monitoring capabilities and immediate feedback for aqua farmers. Despite the widespread use of IoT in many studies [10,15,20,25,28], there is a discernible need for more comprehensive systems that amalgamate IoT with advanced machine learning techniques, ensuring more accurate predictions and decision making. The identified research gaps underscore critical areas for further exploration and development. Firstly, there is a pressing need for the integration of intelligent decision support systems into water quality monitoring platforms, ensuring that the collected data lead to actionable recommendations for farmers in real time. A second crucial research gap involves the development of algorithms for fish species recommendations based on water quality indicators. Leveraging machine learning models to analyze real-time data could significantly enhance the precision of fish species selection, thereby contributing to the overall efficiency and productivity of aquaculture operations. Another notable research gap lies in the creation of real-time predictive analytics for water quality, ensuring that systems can not only analyze historical data but also predict and address potential issues as they occur. There is also a dearth of studies focusing on the development of solar-powered automated self-avoidance robotic agents that can traverse the water surface, collect necessary parameters, and transmit them to a server for further analysis and decision making. Lastly, designing user-friendly interfaces, including mobile applications, is essential in effectively engaging fish farmers, especially those with limited technical expertise. Incorporating farmer feedback into the design process can lead to the development of more user-centric and impactful systems. Addressing these research gaps is imperative for the advancement of intelligent, user-friendly, and sustainable smart pond water quality monitoring and fish farming recommendation systems.

## 3. Materials and Methods

Our proposed integrated System Architecture for the Proposed Model is divided into several parts:

(i) Mobile Robotic Agent: The wheeled robot collects sensor data around the pond. (ii) Fixed Base Unit: It receives data from the mobile robotic agent and sends it to the cloud. (iii) Cloud-Based Data Storage: Central repository for collected sensor data. (iv) Remote Server and Portable Data Monitoring Display: Devices for data retrieval and real-time monitoring. (v) Cloud-Based Data Processing: The collected data on the server undergoes analysis using machine learning algorithms. (vi) Fish Recommendations: Farmers receive suitable fish recommendations using analysed data. Figure 1 depicts a schematic representation of the entire system, which offers an overview of the high-level functional components and how they are interrelated. The flow of information and control inside the system is annotated in the diagram. Each subsystem is elaborately discussed in the following subsections.

### 3.1. Water Quality Parameters

In this study, we have considered three important parameters of water quality monitoring. The parameters are pH, temperature, and turbidity. The standard values of the water quality parameters are demonstrated in Table 1 [1]. These parameters help to determine whether the water quality is suitable for fish farming or not and also to determine the categories of fish that will be ideal to cultivate in certain types of water. Abnormal values of the parameters can greatly affect the production of the fish, as well as being harmful to the aquaculture system [1].

We have considered the temperature as our first parameter because the values of other parameters, such as DO, conductivity, and salinity, directly align with the temperature. Generally, the temperature of the water is identical to the body temperature of fish and it increases or decreases proportionately [30]. A sudden change in temperature, even if it is 5 degrees Celsius, is responsible for the stress and sometimes death of fish. In winter, many fish die because of cold water.

The second parameter considered is the pH, which is important in any aquaculture system since it controls how acidic or alkaline the soil or water is. The pH sensor’s reading ranges from 0 to 14. High or low pH levels over a time period limit fish reproduction. The pH range for warm water is 6.5 to 9, whereas 4 suggests severely acidic water, which causes the death of fish. A score greater than 11 indicates highly alkaline water, which is also responsible for fish mortality. Fish growth is slowed when the pH is between 5 and 6.5 or 9 and 10. When the value is between 4 and 5, there is no reproduction. As a result, it is critical to maintain pH control since it aids in the reduction of other potentially hazardous compounds such as $H_{2}S$ and ammonia gas.

Turbidity, another key water quality indicator, was assessed in this work. A turbidity sensor will aid in determining how much debris is present in the water, since an increase in waste or dust will have an impact on the fish’s survival. Turbidity levels provide a vivid picture of water contamination. Higher turbidity has an impact on aquatic life because it reduces light penetration, damages habitats, and interferes with photosynthesis. Furthermore, it is responsible for creating an appropriate habitat for bacteria and metals that are detrimental to fish. Algae, clay particles, plankton, very small inorganic and organic materials, silt, and other factors contribute to the contamination of water.

Maintaining optimal water quality in fish farming ensures healthy fish growth and prevents disease outbreaks. However, environmental conditions like weather can significantly impact water quality parameters like the pH, temperature, and turbidity. Heavy rainfall can wash in acidic runoff from surrounding land, lowering the pH and potentially stressing fish. Conversely, algal blooms during hot weather can raise the pH, impacting the oxygen availability. Sudden drops in temperature due to storms can shock fish, while prolonged heat waves can reduce the dissolved oxygen levels, creating stressful conditions. Increased erosion from heavy rains can cause sediment runoff, clouding the water and reducing light penetration, hindering photosynthesis and impacting fish feeding. However, real-time monitoring of these parameters, such as the temperature’s delicate balance, the pH’s crucial stability, and the turbidity’s rapid fluctuations, enables the minimization of fish stress, optimization of resource use, and cultivation of a more sustainable, profitable fish farm. Therefore, farmers are recommended to manually adjust the environmental factors after retrieving the values of the parameters using the system.

### 3.2. Mobile Robotic Agent Design Subsystem

We have designed and constructed a lightweight Bluetooth-controlled, obstacle-avoiding robot that can autonomously navigate and collect data across the pond. An SG90 servo motor, a DC motor, an HC-05 Bluetooth module, an Arduino Uno, wheels, and an HC-SR04 ultrasonic sensor are used to construct the robot. The robot detects obstacles in front of it, making navigation easier. Figure 2 depicts the flowchart of the mobile robotic agent.

The robot operates by sending ultrasonic waves from its trigger pin and then receiving ultrasonic signals reflected off an object through its echo pin. This sensor has a range of 2 cm to 400 cm, with accuracy of 3 mm, and can detect obstructions from a 30-degree angle. We utilized a servo motor to move the ultrasonic sensor linked to it to identify the obstruction from every angle. The motor’s shaft rotates from 0 to 180 degrees. The DC motor is used to move the robot with the aid of the L298D motor driver. The motor driver functions as a current amplifier and is used to power DC motors in either direction. Furthermore, users may manually guide the robot forward, backward, left, or right using the ‘Pathfinder’ smartphone application. This communication is carried out using the HC-05 Bluetooth module. In auto mode, after estimating the distance with an ultrasonic sensor, the choice is made to turn left, right, forward, or backward. If the distance between an obstruction and the servo motor is less than 25 cm, the servo motor travels up to 0 degrees to the left and 180 degrees to the right. It rotates on the side with the greatest distance on the left or right. In this manner, the robot is always moving forward. Finally, we utilized solar power as the system’s power source. A battery is linked to it to store the electricity needed to run the system. Furthermore, the panel moves to the sunlight automatically by sensing the angle of the sun with a light-dependent resistor (LDR) and a servo motor. The water quality parameters may be measured from anywhere in the pond using the intelligent obstacle-avoiding robot.

### 3.3. Data Acquisition, Transmission, and  Monitoring Subsystem

In our system, we adopted a pH sensor (Diymore PH-4502C), a turbidity sensor (TSW-20M), and a temperature sensor (DS18B20), as well as a microcontroller-based Arduino Uno board and an ESP32 liquid crystal display for data collection, processing, transmission, and monitoring. Figure 3 depicts the sequence of processes involved in this subsystem.

The first step in the flowchart is system startup, which involves turning on the robot and sensors and programming the microcontroller to read the sensor data. The sensors attached to the robot are used to monitor the water quality parameters in the second stage. The water’s pH, turbidity, and temperature all are monitored. In this work, the water temperature is measured using a waterproof DS18B20 temperature sensor. The TSW-20 M turbidity sensor monitors the water clarity. The pH-4502C sensor measures the hydrogen potential, with recorded values ranging from 0 to 14. In the third stage, the microcontroller ATmega 328p processes the sensor data. The microprocessor translates the raw sensor data into a wirelessly transmittable format. The fourth stage involves wirelessly connecting the microcontroller to the ESP32 module. The ESP32 module is responsible for the wireless transmission of data to cloud-based services (Google Sheets and Firebase) for analysis and visualization. Besides receiving and delivering them, the ESP32 module also processes the acquired data. The sixth step is the analysis of the data stored in the cloud-based platform. Google Sheets is used as a cloud-based database to store the data collected by the robot’s sensors. The data are automatically updated each time the robot sends new data, making it a real-time database for water parameter measurements. This enables the opportunity to analyze the stored data. Real-time data analysis and visualization can be done using Firebase, while machine learning algorithms are applied to the data stored in Google Sheets to generate recommendations for fish cultivation practices. The final step in the flowchart is the retrieval of the data by the user. The user can access the data stored in the cloud-based platform from anywhere in the world, using a web browser or mobile device.

In addition to the cloud-based monitoring system, the system also includes a portable remote monitoring device that provides a real-time display of the quality measurement parameters of the pond water. The display device is designed to be rechargeable and wireless, making it easy to move around and place in different locations within the pond. The device is connected to an ESP32 microcontroller, which reads the data from the cloud-based Firebase database. An I2C display is used to show the values of the different parameters, including pH, temperature, and turbidity, in real time. With this remote monitoring device, farmers can quickly and easily check the water quality at different locations within the pond, without having to access the cloud-based dashboard. This adds a level of convenience and flexibility to the monitoring system and helps farmers to make informed decisions about fish cultivation.

The circuit diagram of the work is depicted in Figure 4. Here, we have used Arduino IDE 2.3.2. as a software tool to program the microcontroller. Arduino IDE has several features that provide an easy-to-use interface to write, edit, debug, and upload programs to Arduino-compatible boards. The language used for this software is C++ or embedded C. We have performed all the coding parts of our project using Arduino IDE.

### 3.4. Fish Recommender Subsystem

In this part, we trained different ML models after collecting and processing the dataset for further analysis.

The conceptual diagram of the whole procedure is demonstrated in Figure 5.

#### 3.4.1. Dataset Collection

The real-time pond water dataset for fish farming [31] utilized in this study was collected from Kaggle https://www.kaggle.com/datasets/monirmukul/realtime-pond-water-dataset-for-fish-farming (accessed on 16 November 2023). The dataset was gathered by the Faculty of Fisheries, University of Dhaka, Dhaka, Bangladesh. The dataset consists of 591 samples and 4 features. The independent variables are pH, temperature, and turbidity, reflecting the measured water quality characteristics. The pH variable has 91 unique values, the temperature variable has 51 distinct values, and the turbidity variable has 108 distinct values. The amount of unique values in each column implies a large range of variability in the examined water quality parameters, which may have substantial consequences for the farming appropriateness of various fish species. On the other hand, the dependent variable is fish, which indicates the fish species being observed. The class variable has 11 distinct fish species, such as katla, song, prawn, rui, koi, pangas, tilapia, silver cup, karpio, magur, and shrimp. The number of fish in each category presented in the dataset is shown in Table 2.

#### 3.4.2. Data Preprocessing

In this section, we checked whether the data were clean, consistent, and acceptable for analysis. We started by searching the dataset for missing values and duplicates. We identified no missing data or duplicates. To mitigate the bias over the target variable, we used a feature transformation technique and a data balancing method.

a:Feature Transformation

It is necessary to convert each feature to the same scale to reduce biases in the dataset. This makes the features of the dataset invariant to the unit. We utilized the standard scaling technique to perform feature scaling to ensure that the input features were all on the same scale. This transformation normalized the input characteristics to have a mean of zero and a standard deviation of one. This enables the faster convergence of ML algorithms. The standard score of a sample *R* of original feature *S* is derived using the formula shown in Equation (Equation 1).
(1)R=(S−m)/d

In Equation (Equation 1), *m* and *d* denote the mean and variance of the training samples, respectively.

b:Solving Class Imbalance Problem

We observed that the records of each fish category were not equal. Thus, we rectified the class imbalance in the target variable by oversampling minority classes using the Synthetic Minority Oversampling Method (SMOTE) [32]. This strategy generates synthetic instances for minority classes, which aids in balancing the distribution of classes in the dataset. After implementing this method, the number of records in each category was 129. The distribution of fish species is demonstrated in Figure 6.

#### 3.4.3. Machine Learning Algorithm

In this study, we have applied eight machine learning models, which are RF, SVM, DT, KNN, logistic regression, bagging, boosting, and stacking. Additionally, we introduce a novel ensemble model as part of our proposed approach, aiming to harness the collective strengths of these base models for improved predictive accuracy and robustness.

Random Forest: We built the model using 50 decision trees and 42 random states. The classifier generates a consensus result from the decision trees.Support Vector Machine: This model used a C = 10 regularization parameter and a gamma value of 0.1. We also set a random state of 42 for consistency.Decision Tree: The maximum depth of the model was set to 5 by us, and we also set a random state of 42.K-Nearest Neighbors: In KNN, k = 5 demonstrated better results.Bagging: The bagging classifier used the DT as its basis estimator and trained 50 different copies of the model to generate a superior ensemble model with the random state 42.Boosting: The AdaBoost classifier utilized the DT as the basic estimator and repeatedly trained many weak models to generate a superior classification model. We have utilized n_estimators 50 and random state 42 for this task.Stacking: The stacking classifier integrated numerous estimators (random forest, SVM, decision tree, and KNN) to produce a meta-model that predicted the output of each base estimator. We set the final estimator to the logistic regression classifier.Logistic Regression: The model was tuned using GridSearchCV with a parameter grid encompassing penalty terms (L1, L2), regularization strengths (C values), intercept fitting options (fit_intercept), solvers (liblinear, saga), and maximum iterations (max_iter). The optimal configuration was determined through a 5-fold cross-validated search, leveraging parallel processing. The resulting model encapsulated the best hyperparameter set for enhanced predictive performance.Proposed Ensemble Model: Our proposed ensemble model combines three ensemble models, bagging, boosting, and stacking. It includes a bagging classifier based on a tuned RF as the base estimator using random state 42; a stacking classifier integrating the tuned RF, tuned DT, tuned AdaBoosting and tuned gradient boosting; and a boosting classifier using the optimized RF. We used the soft voting technique to predict the output by merging bagging, stacking, and boosting. The algorithm of our proposed ensemble model is shown in Algorithm 1.

**Algorithm 1** Proposed Custom Ensemble Method
1:**Input:**$D_{r}$ = Dataset; n_models = Number of base models;2:**Output:** Trained ensemble model merging bagging, boosting, and stacking.3:**function** Ensemble Pseudocode($D_{r}$,n_models)4:    **Initialization:**5:    Set n_models to the desired number of base models.6:    Initialize an empty list model_list to store the trained base models.7:    **for** i=1 to n_models **do**8:        **Training the Base Model:**9:        Train a base model to predict class labels on the entire training dataset $D_{r}$.10:       Append the trained model to model_list.11:    **end for**12:    **Making Predictions:**13:    Initialize an empty list predictions to store predictions from each base model.14:    **for** each base model in model_list **do**15:        Use the base model to predict the class labels for the testing dataset.16:        Append the predictions to predictions.17:    **end for**18:    **Combining Predictions:**19:    For each instance in the testing dataset:20:       Count the class labels predicted by each base model.21:       Select the most frequent class label as the final prediction.22:    **Evaluating Ensemble Performance:**23:    Evaluate the ensemble’s performance on the test dataset.24:
**end function**



### 3.5. Web Interface Design

First, we designed a web form with HTML, employing CSS to produce a seamless user experience with a contemporary visual appearance. The web form was designed for the recording of the water quality parameter data from users. We utilized the Flask web framework to construct this web application. The trained machine learning model was loaded onto the web.

## 4. Results

### 4.1. Performance Evaluation Metrics

Several essential metrics are used to assess the performance of classification models. The confusion matrix, a table that compares model predictions to actual labels, is one such statistic that is crucial in measuring a model’s performance. The four key components of the confusion matrix are true positive (TP), false positive (FP), true negative (TN), and false negative (FN). The number of positive occurrences successfully predicted by the model is marked by TP, whereas the number of negative instances incorrectly projected as positive is denoted by FP. TN, on the other hand, represents the number of negative occurrences that were correctly anticipated as negative, and FN denotes the number of positive instances that were incorrectly classified as negative. Several metrics derived from the confusion matrix assist in assessing the model’s performance.

Accuracy is a statistic that measures the model’s predictability by dividing the total number of occurrences by the sum of TP and TN, as indicated in Equation (Equation 2).
(2)$$Accuracy=\frac{TP}{TP+TN+FP+FN}$$

Precision, a positive predictive value determined using Equation (Equation 3), is concerned with the fraction of correctly predicted positive occurrences out of all cases anticipated.
(3)$$Precision=\frac{TP}{TP+FP}$$

In contrast, recall, as stated in Equation (Equation 4), is the proportion of correctly predicted positive occurrences relative to all positive instances. It is also known as sensitivity or the true positive rate.
(4)$$Recall=\frac{TP}{TP+FN}$$

The F1-scoregives a balanced statistic by assessing the harmonic mean of the accuracy and recall. Equation (Equation 5) may be used to obtain the F1-score. It demonstrates the model’s capacity to achieve both accuracy and recall simultaneously.
(5)$$F1-Score=2*\frac{Precision*Recall}{Precision+Recall}$$

Moreover, the Matthews correlation coefficient, a statistic, considers the confusion matrix’s TP, TN, FP, and FN components. Its value ranges from −1 to 1, with 1 representing an error-free classification, 0 representing an arbitrary classification, and −1 representing a completely incorrect classification. The MCC formula incorporates TP, TN, FP, and FN to provide a comprehensive measure of the categorization quality, as shown in Equation (Equation 6).
(6)$$MCC=\frac{\left(TP*TN)-(FP*FN\right)}{\sqrt{\left(TP+FP)*(TP+FN)*(TN+FP)*(TN+FN\right)}}$$

The ROC AUC metric is used to compare machine learning models. The ROC curve illustrates how well a binary classifier performs at various categorization levels. At different threshold values, it shows the true positive rate (TPR) vs. the false positive rate (FPR). A multiclass ROC curve is utilized in this comparison to assess how well each method performed for each class concerning the other classes. The Area Under the Curve (AUC) score gauges the algorithm’s performance for each class. The range of the AUC is (0,0 to 1,1). The highest value of the AUC indicates a better classifier.

### 4.2. Experimental Design of the Proposed System

The robot designed for this work is capable of moving around the pond, collecting and sending real-time data. The prototype of the system is shown in Figure 7a. We have deployed our system on a pond in Kaliakair, Gazipur, Bangladesh, and collected the parameter values. The values of the parameters are also shown on the LCD display (see Figure 7b. However, the recommender system is implemented on Google Colab using a Linux OS.

### 4.3. Machine Learning Model Result Comparison

We have applied nine ML algorithms to perform the prediction of suitable fish for individual ponds based on several parameters, such as the accuracy, precision, recall, F1-score, and ROC curve. We first evaluated the performance of these models without balancing the dataset. After this, we also evaluated the models after applying the balancing technique. The confusion matrix of the proposed ensemble model before and after applying SMOTE is shown in Figure 8.

The performance of the model is shown in Table 3. The table shows the results of the models both without and after applying SMOTE. Without SMOTE, the RF model exhibited the best results (85% accuracy, 88% precision, 85% recall, 86% F1-score, and 83% MCC). The proposed ensemble model ranks second in terms of accuracy (83%), precision (86%), recall (83%), F1-score (84%), and MCC (81%). The performance of boosting is followed by that of bagging. Stacking also performs in almost the same manner as boosting. Several models, such as LR, SVM, KNN, and DT, did not demonstrate better results in our case. The performance of LR deteriorates because of the inherent linearity assumption, making it less adept in capturing complex, non-linear relationships in the dataset. The test speed analysis shows that DT is the fastest model, with a test speed of 0.00371 s, making it ideal for real-time applications. LR and KNN also perform well with speeds of 0.01960 and 0.02989 s, respectively. RF and bagging balance speed and accuracy at 0.12609 and 0.16518 s, while SVM falls within this range at 0.18323 s. Boosting, stacking, and the proposed ensemble model exhibit slower speeds at 0.18795, 0.69595, and 1.22476 s, respectively.

After applying SMOTE, it is evident that the ensemble model achieved the highest performance, with 94% accuracy, 94% precision, 94% recall, a 94% F1-score, and a 93% MCC. RF showed the second highest performance, with 93% accuracy, 94% precision, 93% recall, a 93% F1-score, and a 93% MCC. On the other hand, LR showed the worst accuracy (51%), precision (49%), recall (51%), and F1-score (49%). The bagging and boosting models exhibited similar performance to RF and the same accuracy, precision, and recall, with a score of 93% and a 92% MCC score. The stacking model showed similar results to the bagging and boosting models, but the MCC score was relatively lower than that of these two ensemble methods, with a score of 47%. Followed by the three ensemble classifiers, KNN also showed good performance, with 88% accuracy, 89% precision, 88% recall, and an 88% F1-score. The performance of DT was worse than that of KNN. SVM exhibited the second-lowest performance in recommending appropriate fish for farming in terms of accuracy, precision, recall, and the F1-score. The ROC curve is another metric used to compare the performance of models. We have used the OnevsRest method to draw the ROC curve for multi-class classification. From the curves demonstrated in Figure 9 and Figure 10, we can conclude that, even without balancing the dataset, the ensemble model performs best. The AUC score of the ensemble model is between 0.89 and 1.00 before applying SMOTE and between 0.99 and 1.00 after applying SMOTE when differentiating the classes. It can also be concluded from the analysis of the AUC score that LR is the worst model in differentiating classes. The proposed ensemble model exhibits the lowest performance in the case of the test speed, which is 1.40 in seconds after SMOTE. The results of the evaluation show that our proposed ensemble model is the most effective model in solving this multiclass classification issue.

### 4.4. Result Interpretation of the Models

Shapley Additive exPlanations (SHAP) values provide insights into the importance of individual features in influencing model predictions across all classes. The impact of the parameter on the proposed model’s output is depicted in Figure 11. In the figure, the x-axis represents the three pond water properties influencing fish recommendations: pH, temperature, and turbidity. The y-axis displays the average SHAP value for each feature, indicating its impact on the proposed model’s predictions. Larger bars signify a more significant influence. Turbidity has the largest, followed by temperature and pH, highlighting their relative importance in determining suitable fish species. Thus, it is evident that variations in turbidity have a substantial effect on fish suitability, probably as a result of species-specific sensitivity. Temperature also plays a significant role, while the pH exerts a moderate influence despite its lower average SHAP value, potentially due to varying species’ pH requirements. These nuanced insights into the feature importance shed light on the inner workings of our model, enabling us to interpret its behavior and paving the way for potential performance improvements.

### 4.5. Fish Recommendation System Web Application

One of the project’s features is a user-friendly web interface. It allows users to provide input values of various parameters related to their pond’s water quality, such as the temperature, pH level, and turbidity. The input is collected through a form that has been integrated into the website. Once the user has submitted their input, the data are processed by the recommender system. The trained ensemble model is deployed on the webpage that predicts the output. Figure 12 shows a snapshot of the web interface and the prediction result.

## 5. Discussion

The originality of this project resides in its potential to merge several cutting-edge technologies to transform traditional fish farming. The architecture of the system is the integration of different components, enabling the efficient collection, processing, and analysis of data. This enables farmers to make the right decisions and optimize their fish cultivation practices for increased productivity with smaller production costs. The main objective of this work was to collect and monitor the water quality of ponds and recommend suitable fish species to cultivate. We have reviewed several papers that have considered the monitoring and control of the parameters of water quality. We have provided a comparative study with these works in Table 4. Most of the works considered pH and temperature values to determine the quality and did not consider the turbidity parameter [10,13,17,25,33,34,35]. We have also utilized turbidity sensor values along with the two sensors, as it is one of the key features. There are some works in which the authors implemented solar power to operate the system, while there exist several works [9,13,17,33,35,36] that did not utilize solar power. One of the unique features of this study is that we have implemented an ML algorithm on a dataset and designed a recommender system that is able to predict fish suitability. Moreover, the recommender system adds further uniqueness to this study, as there is no other work(to our knowledge) that has developed an ML-based web interface for suitable fish recommendations. Additionally, we found no work that has designed a moving robot to collect data from various locations in the pond. Moreover, the cost of the complete system is low, at approximately eight thousand Bangladeshi taka.

Apart from the system design, we have compared the performance of our proposed ML model with [29] as there are no other works (to our knowledge) that have implemented ML algorithms on this dataset for the prediction of suitable fish. However, the authors did not use feature scaling and data balancing techniques. They concluded that the RF model performed best, with accuracy of 88.48%, whereas we have achieved 94% accuracy with our proposed ensemble model. This indicates that feature scaling and data balancing improve the model performance. Furthermore, we have designed a complete IoT system, which is not considered in other works. It can be concluded from Table 5 that our proposed model has achieved superior performance compared to existing works. Our method is reasonably priced since it not only boosts productivity and increases profits but also decreases the duration and intensity of the producer’s manual labor, minimizing the frequency with which farmers must work at night and increasing farmer satisfaction. The system is very scalable in the sense that it can be implemented in multiple ponds simultaneously.

## 6. Conclusions

The developed solar-powered automated robot presents an innovative and cost-effective solution for the monitoring of pond water quality in sustainable fish farming practices. This system has the potential to revolutionize the industry by enabling independent data collection and analysis, leading to informed decision-making for farmers. By incorporating cloud-based technology, the system allows for easy access to and monitoring of data from any location. This widens the visibility and enhances the ability to monitor water quality parameters effectively. Fish farmers stand to benefit greatly from timely interventions to maintain optimal water quality conditions. Moreover, the integration of machine learning algorithms provides an additional advantage by predicting suitable fish species for farming based on the water quality data. This feature further streamlines the decision-making process for farmers, allowing them to choose the most appropriate fish species that thrive in specific water conditions. By deploying this system, a conducive environment for aquatic life can be achieved, resulting in increased productivity and profitability for fish farmers. Furthermore, the system aids in minimizing the daily operating costs, making fish farming more sustainable and economically viable. Overall, the developed solar-powered automated robot, combined with cloud-based monitoring and machine learning algorithms, offers significant benefits for sustainable fish farming. It enhances productivity, reduces costs, and promotes a healthy aquatic environment, leading to increased profitability for fish farmers. Our future research endeavors will delve into three primary areas. Firstly, we aim to optimize the data analysis techniques, ensuring improved accuracy and efficiency in the machine learning algorithms employed for fish species prediction based on water quality. Secondly, we intend to explore the integration of supplementary sensors to monitor a wider range of essential parameters beyond the pH, temperature, and turbidity. This expansion will provide a more comprehensive understanding of the pond environment. Lastly, we aspire to automate water quality management processes, enabling proactive actions to maintain optimal conditions. By pursuing these avenues, we aim to enhance the effectiveness and performance of our solution in promoting and sustaining optimal water quality for fish farming.

## Figures and Tables

**Figure 1 sensors-24-03682-f001:**
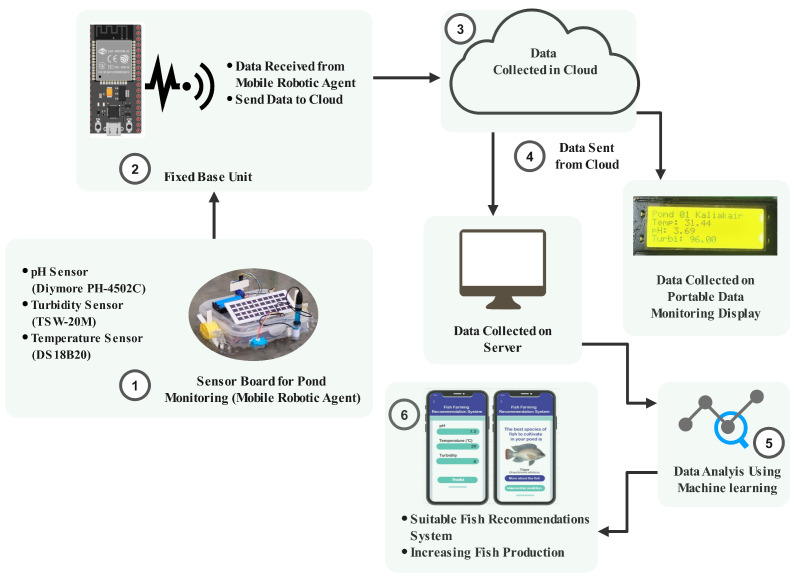
Integrated system architecture for the proposed model.

**Figure 2 sensors-24-03682-f002:**
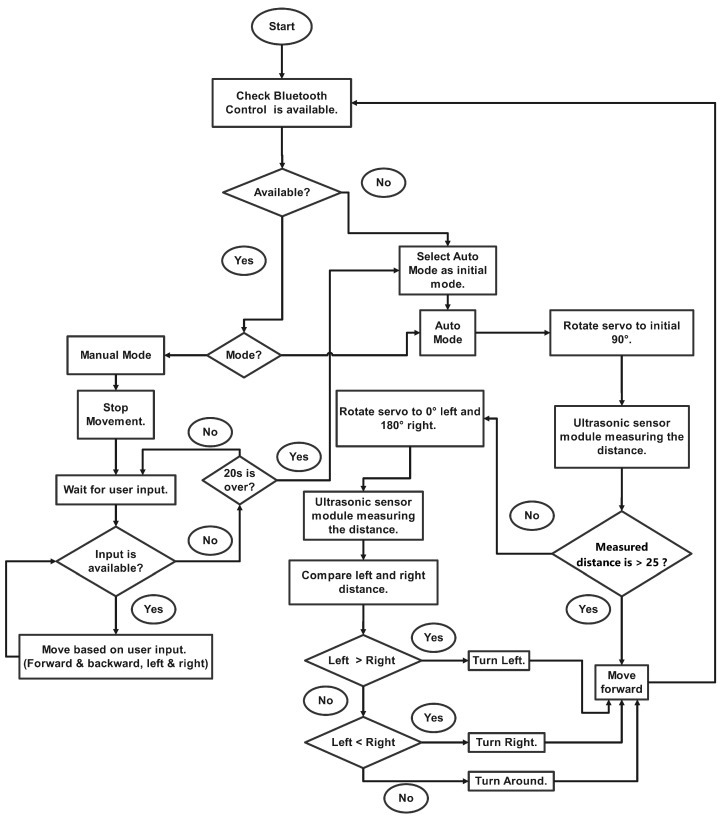
Flowchart of mobile robotic agent.

**Figure 3 sensors-24-03682-f003:**
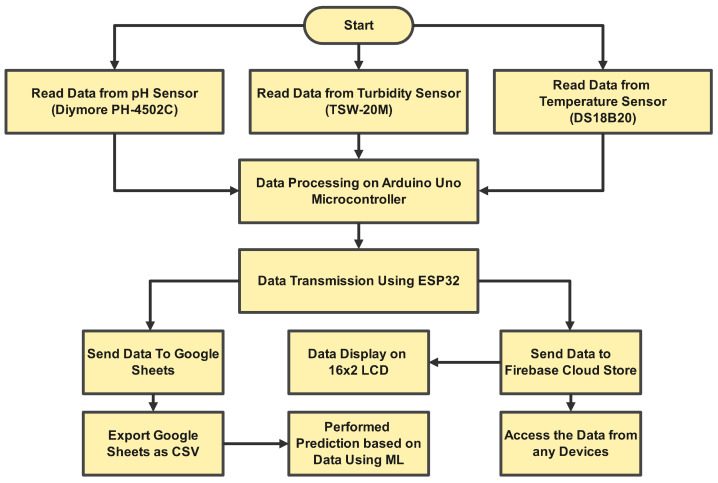
Flowchart of the proposed model.

**Figure 4 sensors-24-03682-f004:**
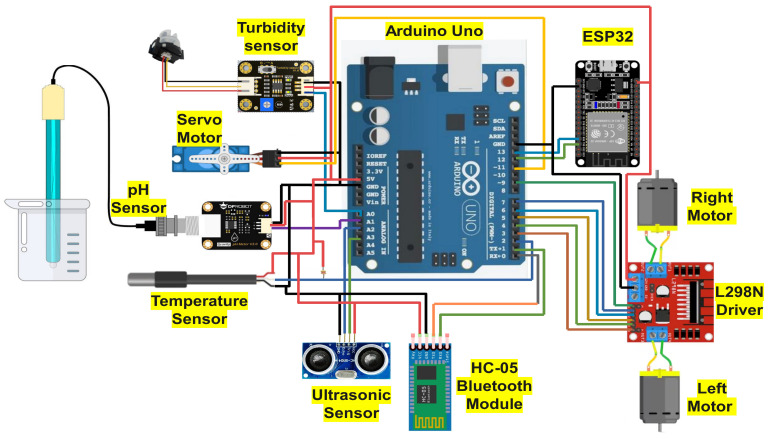
Circuit diagram of the proposed system.

**Figure 5 sensors-24-03682-f005:**
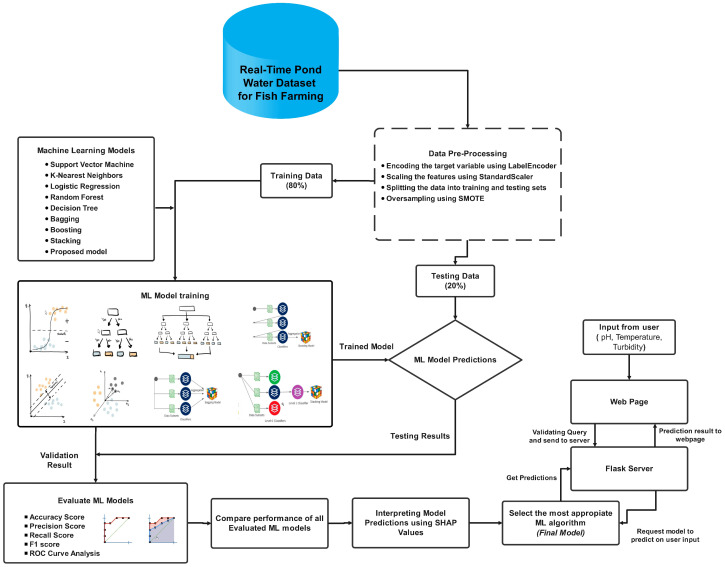
Conceptual diagram of ML model.

**Figure 6 sensors-24-03682-f006:**
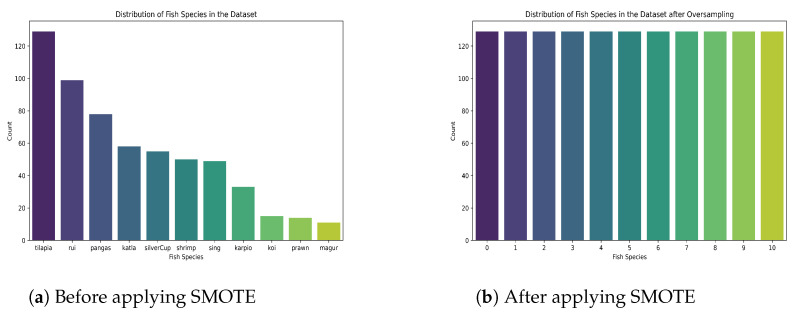
Distribution of fish species before and after applying SMOTE.

**Figure 7 sensors-24-03682-f007:**
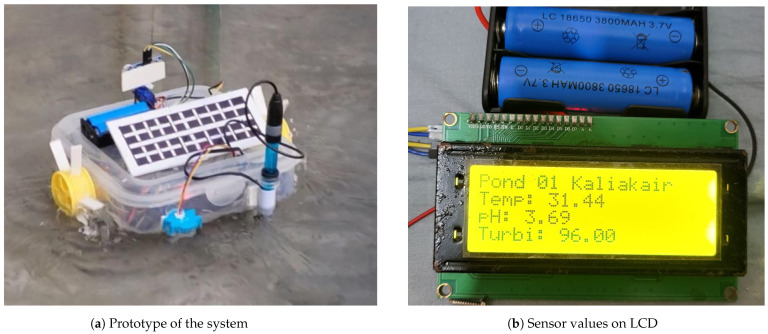
System prototype and portable display system of the prototype.

**Figure 8 sensors-24-03682-f008:**
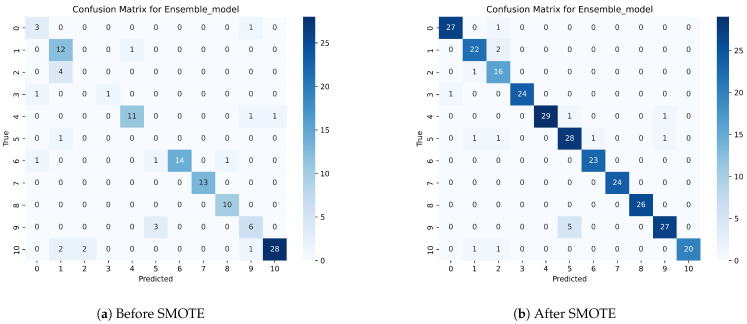
Confusion matrix of ensemble model before and after SMOTE.

**Figure 9 sensors-24-03682-f009:**
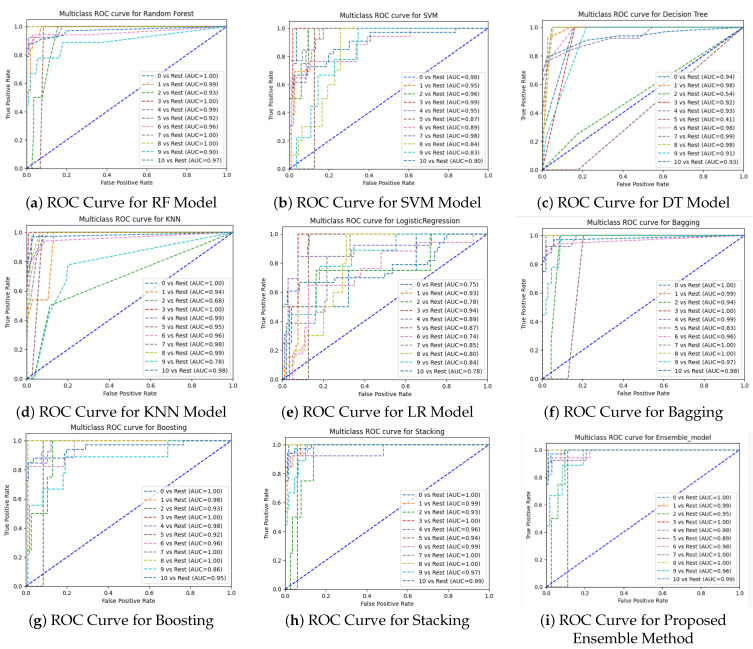
ROC curves of different ML algorithms before applying SMOTE.

**Figure 10 sensors-24-03682-f010:**
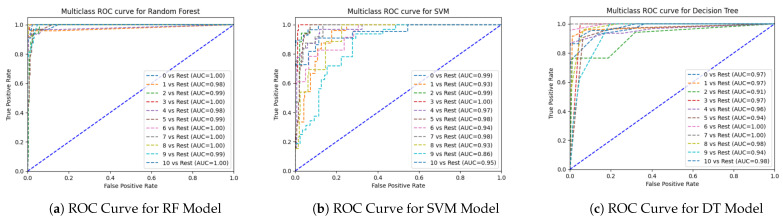
ROC curves of different ML algorithms after applying SMOTE.

**Figure 11 sensors-24-03682-f011:**
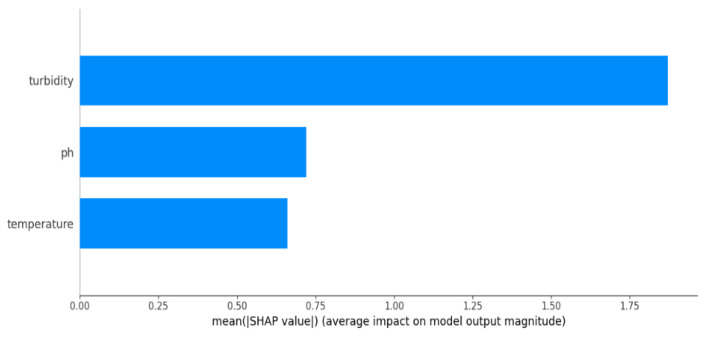
Influence of the parameters on the output using SHAP.

**Figure 12 sensors-24-03682-f012:**
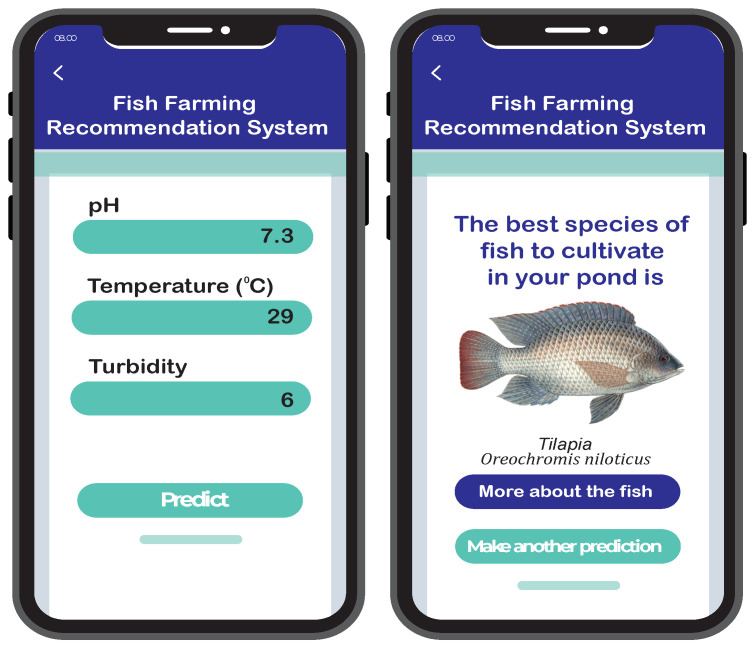
Fish Recommender System.

**Table 1 sensors-24-03682-t001:** Reference values of each water quality parameter.

Water Quality Parameter	Value
Temperature	25 °C–32 °C or >20 °C
pH	6.5–8.5
Turbidity	30–80 cm
Dissolved Oxygen (DO)	>5 mg/L
Biochemical Organic Demand (BOD)	<5 mg/L
Total Dissolved Solids (TDS)	400 mg/L
Chemical Oxygen Demand (COD)	20–30 mg/L
Total Suspended Solids (TSS)	<80 mg/L
Electrical Conductivity (EC)	150–50 micros/cm
Hardness	>15 mg/L
Alkalinity	50–300 mg/L
Nitrite NO_2_	<0.2
Nitrate NO_3_	0–100
Total Ammonia Nitrogen (TAN)	0–0.2

**Table 2 sensors-24-03682-t002:** Number of fish in each species.

Class Label	Tilapia	Rui	Pangas	Katla	Silver Cup	Shrimp	Sing	Karpio	Koi	Prawn	Magur
**Number**	129	99	78	58	55	50	49	33	15	14	11

**Table 3 sensors-24-03682-t003:** Performance comparison among various ML algorithms.

	Without SMOTE						After SMOTE					
	Acc	Pre	Rec	F1	MCC	TS	Acc	Pre	Rec	F1	MCC	TS
**Ensemble Model**	0.83	**0.86**	**0.83**	**0.84**	**0.81**	1.22476	**0.94**	**0.94**	**0.94**	**0.94**	**0.93**	1.40498
**RF**	**0.85**	**0.88**	**0.85**	**0.86**	**0.83**	0.12609	**0.93**	**0.94**	**0.93**	**0.93**	**0.93**	0.01370
**SVM**	0.45	0.55	0.45	0.45	0.37	0.18323	0.64	0.63	0.64	0.6	0.62	0.01291
**DT**	0.71	0.78	0.71	0.72	0.68	0.00371	0.82	0.84	0.82	0.81	0.80	0.00020
**KNN**	0.66	0.74	0.66	0.68	0.61	0.02989	0.88	0.89	0.88	0.88	0.87	0.01352
**LR**	0.38	0.35	0.38	0.34	0.80	0.0196	0.51	0.49	0.51	0.49	0.91	0.00039
**Bagging**	0.82	0.85	0.82	0.83	0.79	0.16518	0.93	0.93	0.93	0.93	0.92	0.47181
**Boosting**	0.82	0.84	0.82	0.82	0.80	0.18795	0.93	0.93	0.93	0.93	0.92	0.50629
**Stacking**	0.82	0.81	0.82	0.81	0.27	0.69595	0.93	0.93	0.93	0.93	0.47	0.04793

**Table 4 sensors-24-03682-t004:** Comparative analysis of the proposed system with existing state-of-the-art systems.

Refs.	Number of Sensors Used	Parameters Considered <pH, Temperature, Turbidity> <yes/no, yes/no, yes/no>	Solar Power	Data Collection Robot	Cloud Platform	Fish Recommender	Productivity Concern	Cost–Benefit Concern
[10]	4	yes, yes, no	yes	no	yes	no	yes	no
[33]	5	yes, yes, yes	no	no	no	no	yes	yes
[25]	4	yes, yes, no	yes	no	yes	no	no	yes
[34]	4	yes, yes, no	yes	no	no	no	yes	yes
[13]	4	yes, yes, no	no	no	yes	no	no	no
[36]	4	yes, yes, yes	no	no	yes	no	yes	no
[35]	4	yes, yes, no	no	no	yes	no	yes	yes
[17]	4	yes, yes, no	no	no	yes	no	yes	yes
[9]	5	yes, yes, yes	no	no	yes	no	no	yes
[37]	2	yes, yes, yes	yes	no	yes	no	yes	yes
[38]	2	yes, yes, no	yes	no	yes	no	yes	no
[39]	2	yes, yes, no	no	no	yes	no	yes	yes
[40]	2	yes, yes, no	no	no	yes	no	no	no
[41]	2	yes, yes, no	no	no	yes	no	yes	yes
[42]	3	yes, yes, yes	no	no	yes	no	yes	yes
[29]	2	yes, yes, no	no	no	yes	no	yes	yes
[6]	2	yes, yes, no	no	no	yes	no	yes	no
**Proposed System**	**3**	**yes, yes, yes**	**yes**	**yes**	**yes**	**yes**	**yes**	**yes**

**Table 5 sensors-24-03682-t005:** Comparative analysis of ML model performance with existing works.

Reference	Utilized Model	Best Model	Tools	Preprocessing: Feature Scaling, SMOTE	Performance Metric	Best Model Accuracy
Islam et al. [29]	RF, J48, NB, KNN, CART	RF	Weka	No	Accuracy, Kappa, TPR	88.48
Proposed	Ensemble, RF, DT, NB, KNN, SVM, LR, Bagging, Boosting, Stacking	Ensemble	Google Colab	Yes	Accuracy, Precision, Recall, F1-score, ROC AUC	94

## Data Availability

Data are contained within the article.

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
