# Peer review of "An Integrated Smart Pond Water Quality Monitoring and Fish Farming Recommendation Aquabot System"

_sensors, 2024, doi:10.3390/s24113682_

Round 1

Reviewer 1 Report

Comments and Suggestions for Authors

In this paper, a system can detect the pH value, temperature and turbidity in the pond through mobile robots, and recommend the most suitable fish for aquaculture according to the water quality index, so as to help farmers make breeding decisions. The proposed water quality testing system has a high classification detection rate and can be deployed on the web side. It is easy to use and has practical application value. Overall, the article demonstrates a level of innovation.  

1.The third line of the second section has an extra "?" in the citation format.

 2.There is a gap in the second line of Section 3.1 where [] is quoted.

 3.Can the classification effects of previous studies, such as recognition rate, be included in Section 2?

 4.In order to demonstrate the practical value of the system, please explain whether environmental conditions such as weather will have a large impact on water quality and whether system decisions in the same water area will change in the short term.

 5.Water quality is determined by a variety of factors, in order to improve the robustness of the results, please further explain the determination factors of PH, temperature, and turbidity for farmed fish through research or literature.

 6.The analysis of the results is relatively simple, and it is recommended to further analyze the influence of pH value, temperature and turbidity on the classification results.

 7.In Section 3.5.3, the LR model has the worst recognition results, please briefly explain why.

 8.It is suggested that the confusion matrix obtained by the optimal model test should be displayed in the result analysis to improve the intuitiveness of the results.

 9.The description of the dataset is lacking in this paper, and it is recommended to use a chart to introduce the content and size of the dataset.

 10. In order to prove that the network model in this paper is superior to other models, the test speed should be added, and the detection efficiency of each network should be analyzed and compared in the following section.

Reviewer 2 Report

Comments and Suggestions for Authors

1. please check the references, such as not annotated, wrong order or wrong format.

2. the chart should be placed before and after the corresponding paragraph, not inserted in the middle of the paragraph, not to mention in other chapters.

3. Figure 2 is not quoted in the paper, and Figure 4 should be clearly commented in the paper.

4. The meaning of each letter in the formula must be explained.

5. The content of section 3.4.2 is not clear, so numbers can be marked to explain separately.

6. Don't add any punctuation in the title

7. It is best not to use the same serial number format as the title in section 3.4.3.

8. The font size of formula 6 is inconsistent with other formulas.

9. Section 3.5.2 shows that systematic analysis has been carried out in this section, but there is no relevant content. Check whether the structure of the article is wrong.

10, the caption of the picture should be centered under the picture, please check this part of the article.

11, Table 4 format is not correct, the title line should be separated from the content with a horizontal line.

12. There is too much white space in Figure 1.

13. Please add the robot model diagram designed in section 3.2 and the dataset sample diagram in section 3.4.1.

14. Use high-resolution images.

15, the machine learning algorithm is built on which platform, please specify the experiment related environment.

Reviewer 3 Report

Comments and Suggestions for Authors

AquaBot: A Smart Pond Water Quality Monitoring and Fish Farming Recommendation System is presented in this work. Following are the comments for the improvement of this article

The main contributions of this work is well presented, however, the novelty of this work can be highlighted in this.

Figure 1 proposed model does not look professional and that can be redrawn neatly.

Why only 591 samples are considered? More samples can't be generated from this model?

Is it a regression based problem or classification based problem?

How multiclass problem handled with linear regression?

How 9 already existing approaches are claimed as proposed model?

The authors should find a single model as a proposed model through their experimentation.

This paper should be rewritten on the given points and the appropriate model should be proposed.

Round 2

Reviewer 1 Report

Comments and Suggestions for Authors

According to the comments, the authors revised the paper, added the overall experimental process, described the experimental process in more detail, improved the charts, and further described the water quality parameters, and found no major problems after the re-evaluation. Overall, this paper demonstrates a certain level of innovation in the field of water quality testing and aquaculture.  

Author Response

Thank you so much for your efforts and nice comment.

Reviewer 3 Report

Comments and Suggestions for Authors

Even though the majority of the comments are addressed, the newly included ensemble model should be elaborated in detail. Pseudocode or algorithm can be included to understand how this proposed algorithm works.

Author Response

Thank you so much for suggestion. We have added the algorithm of our proposed ensemble model in the updated manuscript.